# Serum Leptin Levels, Nutritional Status, and the Risk of Healthcare-Associated Infections in Hospitalized Older Adults

**DOI:** 10.3390/nu14010226

**Published:** 2022-01-05

**Authors:** Elena Paillaud, Johanne Poisson, Clemence Granier, Antonin Ginguay, Anne Plonquet, Catherine Conti, Amaury Broussier, Agathe Raynaud-Simon, Sylvie Bastuji-Garin

**Affiliations:** 1Department of Geriatrics, Hôpital Europeen Georges Pompidou, Paris Cancer Institute CARPEM, AP-HP, F-75015 Paris, France; johanne.poisson@aphp.fr (J.P.); catherine.conti@aphp.fr (C.C.); 2INSERM, the University of Paris Est Creteil, IMRB, F-94010 Creteil, France; amaury.broussier@aphp.fr (A.B.); sylvie.bastuji-garin@aphp.fr (S.B.-G.); 3Faculty of Health, University of Paris, F-75006 Paris, France; clemence.granier@aphp.fr (C.G.); agathe.raynaud-simon@aphp.fr (A.R.-S.); 4Biological Immunology Department, Hôpital Europeen Georges Pompidou, Paris Cancer Institute CARPEM, AP-HP, F-75015 Paris, France; 5Service de Biochimie, Hôpital Cochin, AP-HP, F-75014 Paris, France; antonin.ginguay@aphp.fr; 6EA4466, Faculté de Pharmacie, University of Paris, F-75006 Paris, France; 7Department of Biology-Pathology, Hôpitaux Henri-Mondor, AP-HP, F-94010 Creteil, France; anne.plonquet@aphp.fr; 8Department of Geriatrics, Hopitaux Henri-Mondor/Emile Roux, AP-HP, F-94456 Limeil Brevannes, France; 9Department of Geriatric, Bichat, Beaujon and Bretonneau Hospitals, AP-HP, F-75018 Paris, France; 10Department of Public Health, Hopitaux Henri-Mondor, AP-HP, F-941010 Creteil, France

**Keywords:** leptin, hospital-acquired infection, malnutrition, older women

## Abstract

We aimed to determine whether serum leptin levels are predictive of the occurrence of healthcare-associated infections (HAIs) in hospitalized older patients. In a prospective cohort, 232 patients had available data for leptin and were monitored for HAIs for 3 months. Admission data included comorbidities, invasive procedures, the Mini Nutritional Assessment (MNA), BMI, leptin, albumin and C-reactive protein levels, and CD4 and CD8 T-cell counts. Multivariate logistic regression modelling was used to identify predictors of HAIs. Of the 232 patients (median age: 84.8; females: 72.4%), 89 (38.4%) experienced HAIs. The leptin level was associated with the BMI (*p* < 0.0001) and MNA (*p* < 0.0001) categories. Women who experienced HAIs had significantly lower leptin levels than those who did not (5.9 μg/L (2.6–17.7) and 11.8 (4.6–26.3), respectively; *p* = 0.01; odds ratio (OR) (95% confidence interval): 0.67 (0.49–0.90)); no such association was observed for men. In a multivariate analysis of the women, a lower leptin level was significantly associated with HAIs (OR = 0.70 (0.49–0.97)), independently of comorbidities, invasive medical procedures, and immune status. However, leptin was not significantly associated with HAIs after adjustments for malnutrition (*p* = 0.26) or albuminemia (*p* = 0.15)—suggesting that in older women, the association between serum leptin levels and subsequent HAIs is mediated by nutritional status.

## 1. Introduction

The hormone leptin is secreted by adipocytes and regulates food intake and thermogenesis [1]. Serum leptin levels are correlated with fat mass [2], and are thus higher in women than in men. Some studies [3,4,5,6] (but not others [7]) have observed an association between low leptin levels and malnutrition in patients with severe chronic diseases (such as kidney failure and cirrhosis), independently of the level of inflammation or the fluid balance. Furthermore, a growing body of evidence suggests that leptin can also regulate immune function [8] (especially T lymphocyte function [9,10]) and may be involved in the response to infection [11]. However, the value of leptin for predicting healthcare-associated infections (HAIs) in older inpatients remains to be determined.

HAIs are highly prevalent in hospitalized older patients and may be associated with functional decline, high health care costs, and mortality [12,13]. We have reported previously that in older patients admitted to a rehabilitation care facility, subsequent HAIs were associated with lower baseline values for energy intake and serum albumin, independently of immune variables [14]. Other researchers have observed an association between malnutrition and HAIs [15,16,17]. Importantly, the improvement of nutritional care in malnourished in-patients or in-patients at risk of malnutrition resulted in a 35.7% reduction in HAI rate [18].

Hence, we hypothesize that a low leptin level would be associated with a greater risk of HAIs.

The objectives of the present study were therefore to investigate the relationship between leptin levels and nutritional variables in older patients hospitalized in rehabilitation unit, and to determine whether lower leptin levels were associated with HAIs.

## 2. Materials and Methods

### 2.1. Patients and Study Design

We analysed the data collected during a previously described prospective cohort study conducted between July 2006 and December 2008 in a French teaching hospital [19]. The cohort comprised 252 consecutive Caucasian patients aged 75 years or over, and who were referred to the geriatric rehabilitation unit by acute medical or surgical units during the study period. The main inclusion criteria were stable clinical status upon admission and the need for long-term care or rehabilitation. The main exclusion criteria were terminal disease, fever, infection, cancer, or known immunological dysfunction on admission. Furthermore, patients were not included if they spent less than 48 h in the geriatric rehabilitation unit. The study participants were followed up for HAIs for up to 3 months in the rehabilitation unit or until death or discharge from the rehabilitation unit.

Here, 232 patients with leptin assay data at baseline were included in the study. The study was approved by an independent ethics committee (CPP Ile-de-France IX ethics committee, Paris, France; reference: SCR06010). Written informed consent was obtained from each patient before the study inclusion.

### 2.2. Data Collection

Upon admission to the geriatric rehabilitation unit, the following variables were collected prospectively on a standardized case report form by a geriatrician: age, sex, reason for admission, cognitive status (according to the Mini-Mental State Examination score (MMSE) [20]), functional status (according to the six-item Activities of Daily Living (ADL) score [21]), and comorbidities by the Cumulative Illness Rating Scale—Geriatric (CIRS-G) [22]. The MMSE consists of 11 questions, grouped into seven cognitive domains (time, place, three words memory, attention and calculation, language, and visual construction) and is scored on 30 points [20]. The ADL score is a composite score measuring the patient function on six different areas (bathing, dressing, transferring, ambulation, eating, and continency) [21]. The CIRS-G score summarizes disease burden (comorbidity index) across 14 organ-systems, with each item scored 0 to 4 (no to extremely severe problem) [22].

### 2.3. Nutritional Assessment

Body weight, height, and BMI (weight (kg)/height (m^2^)) were recorded. The Mini Nutritional Assessment (MNA) score [23] was also recorded. The MNA^®^ contains 18 items and evaluates four different dimensions: anthropometrics (BMI, weight loss, and arm and calf circumferences), general aspects (lifestyle, medication, mobility, and presence of signs of depression or dementia), diet (number of meals, food and fluid intake, and autonomy of feeding), and a self-assessment of health and nutrition. By summing up the four scores (maximum: 30), older people can be classified into three groups, with the following threshold values: <17 for “malnourished”, 17–23.5 for “at risk of malnutrition” and ≥24 for “normal nutritional status”.

### 2.4. Invasive Procedures

The following invasive procedures were recorded upon admission and during the 3-month follow-up period (or until the occurrence of an HAI, discharge, or death): intravenous catheter placement, indwelling urinary catheter placement, intermittent urinary catheter use, gastrointestinal tract endoscopy, nasogastric tube use, colonoscopy, and bronchoscopy.

### 2.5. Assessment of HAIs

As described previously [19], an HAI was defined as a well-documented infection that was neither present nor incubating upon admission and that met the Centres for Disease Control’s definition of a nosocomial infection [24]. HAIs were diagnosed by consensus between two geriatricians. Once a week for 3 months (or until the occurrence of an HAI, discharge, or death), the two geriatricians visited each study participant and reviewed the latter’s medical records with the attending physician and nurses. The diagnosis was based on a combination of clinical findings (fever, pulmonary rales or dullness, dyspnoea, cough, purulent sputum, dysuria, urgency, suprapubic tenderness, clinical evidence of sepsis, and/or purulent drainage from a surgical incision), laboratory test results (blood and urine cultures, isolation of a pathogen from other specimens, and antigen- or antibody-detection tests), and imaging findings (e.g., X-rays and CT scans). Only HAIs requiring antibiotic therapy were taken into account; asymptomatic urinary tract infections were not included.

### 2.6. Laboratory Variables

All laboratory variables were assessed upon admission to the geriatric rehabilitation unit. Serum leptin levels were assessed using a commercially available ELISA (Human Leptin Quantikine ELISA Kit (R&D Systems, Abingdon, UK)). The serum samples were stored at −20 °C prior to analysis. The serum albumin was measured using a BNII analyser (Siemens, Saint-Denis, France). C-reactive protein (CRP) was measured by immunoturbidimetry using an Advia 1650 analyser (Siemens Healthcare, Saint-Denis, France). An ultrasensitive CRP assay was not available at the time of the study. Immunological variables were assessed using flow cytometry immunophenotyping, as described previously [25]. Absolute peripheral blood CD4 and CD8 T-cell counts were determined using a Cyto-Stat tetraCHROME device (including labelling with CD45, CD3, CD4, and CD8) and acquisition on an FC500 flow cytometer (both from Beckman Coulter, Villepinte, France). Counts of naïve, memory, and terminal effector T-cells were determined as follows: CD8 and CD4 naïve T-cells were defined as CD45RA+CD62L+, peripheral memory T-cells were defined as CD45RA−CD62L−, and terminal effector CD8 T-cells were defined as CD45RA+CD62L−CD28−. The anti-CD62L antibody was obtained from BD Pharmingen (BD Biosciences, San Jose, CA, USA) and the CD4, CD8, and CD28 antibodies (Caltag) came from Life Technologies (Carlsbad, CA, USA).

### 2.7. Data Analysis

Quantitative and qualitative variables were described as the median (interquartile range (IQR)) and the frequency (percentage), respectively. Continuous variables with a skewed distribution were log-transformed; the corresponding odds ratios (ORs) (95% confidence interval (CI)) were quoted for a 1 standard deviation increment or decrement in the log-transformed value.

Relationships were considered between the baseline leptin level on the one hand, and the patients’ initial characteristics and outcomes on the other.

Given that leptin levels can vary by sex, we first used a Kruskal−Wallis test to compare the leptin level distribution in men vs. women. After stratification by sex, we investigated potential associations between the leptin level on one the hand, and other nutritional variables (BMI, MNA score, and the serum albumin level) and immunological variables on the other hand, by applying Spearman’s rank correlation test or the Kruskal−Wallis test, as appropriate. When no significant differences between sexes were observed, sex-adjusted analyses (using quantile regression models) were performed.

The threshold for statistical significance was set to *p* ≤ 0.05, and p values between 0.05 and 0.10 were denoted as indicating a trend. The test results were not adjusted for multiple comparisons. All of the analyses were performed using STATA software (version 15.0, StataCorp, College Station, TX, USA).

#### Independent Relationships between the Baseline Leptin Level and the Occurrence of HAIs

Patients with vs. without HAIs were compared (using a Kruskal−Wallis test or a chi-squared test) with regard to factors known or suspected to be associated with HAI: at least one invasive medical procedure (yes/no), comorbidities (according to the CIRS-G score), functional status (according to the ADL score), inflammation (as assessed by the CRP level), and nutritional and immunological variables [14,15,16,17,19,25,26]. Crude ORs (95% CIs) were estimated using asymptotic logistic regression analyses for variables with *p* < 0.10. Next, the ORs (95%CI) for leptin were sequentially adjusted for variables associated with HAIs (*p* < 0.10). Lastly, after assessing first-order interactions and confounding factors, we built a multivariate model to determine whether the leptin level was associated with HAIs, independently of other predictors. The model’s discriminant ability was assessed using the C-index (95%CI), while the Hosmer−Lemeshow calibration test was used to assess goodness-of-fit. Lastly, we conducted a sensitivity analysis after dichotomizing the leptin level (≤1st quartile vs. >1st quartile).

## 3. Results

### 3.1. Characteristics of the Study Population

The median (range) age was 84.8 (75–101) years, and 168 (72.4%) of the 232 participants were female (Table 1). The median (IQR) CIRS-G and ADL scores were 11 [9,10,11,12,13,14] and 8 [4,5,6,7,8,9,10,11,12], respectively. According to the MNA, 50 participants (21.6%) showed malnutrition and 125 (53.9%) were at risk of malnutrition. The median leptin level for the study population as a whole was 9.3 (3.7–21.6) µg/L.

During follow-up, 89 patients (38.4%) experienced at least one HAI. The most common HAI sites were the respiratory tract (47.2%, 42 out of 89) and the urinary tract (41.6%, 37 out of 89). At the end of the 3-month follow-up period, 30 patients (13.2%) were still hospitalized and 14 (6.1%) had died.

### 3.2. Associations between the Serum Leptin Level and Patient Characteristics

The serum leptin level was significantly higher in women than in men (median (IQR): 10.6 (4–24.4) and 7.3 (2.5–12.6), respectively; *p* = 0.009). It fell significantly with age in women but not in men (Table 2). The serum leptin level was significantly correlated with the serum albumin level in women only (Rho 0.20, *p* = 0.01) and with the naïve CD4 T-cell count in men only (Rho = 0.27, *p* = 0.034). In both sexes, the median leptin level increased significantly with the increasing BMI class, and decreased significantly with the increasing malnutrition class (as assessed by the MNA score). These relationships were also observed in sex-adjusted analyses.

With regard to outcomes, female patients who experienced one or more HAI had significantly lower values of leptin level than female patients who did not (5.9 (2.6–17.7) versus 11.8 (4.6–26.3), respectively; *p* = 0.01); this difference was not observed for the men (7.9 (2.4–12.9) versus 6.4 (2.5–11.9), respectively; *p* = 0.35).

### 3.3. Factors Associated with HAIs

Given that the association between leptin levels and the occurrence of HAI was observed in women only (Table 2), subsequent analyses were restricted to this population (N = 168).

In a univariate analysis, women who experienced an HAI were more likely to be malnourished (MNA score < 17) and to have invasive procedures than women who did not experience an HAI. The women in the HAI group also had (i) significantly lower values for serum leptin, ADL score, serum albumin, and the naïve CD8+ T-cell count, and (ii) significantly higher values for the CIRS-G score, serum CRP, and the memory CD8+ T cell count (Table 3). A non-significant trend was observed for a lower CD4/CD8 ratio and a higher effector CD8 T-cell count in the HAI group. BMI and age were not associated with HAI.

Adjustments for the ADL and CIRS-G scores; invasive procedures; serum CRP; serum albumin; the CD4/CD8 ratio; and the naïve CD8 T-cell, memory CD8 T-cell and effectors CD8 T-cell counts did not substantially change the relationship between the leptin level and the occurrence of HAIs (Figure 1). In contrast, adjustment for the MNA score or the serum albumin level abolished the significant association between leptin and HAIs (OR = 0.82 (0.57–1.16); *p* = 0.26, and OR = 0.77 (0.54–1.09); *p* = 0.15, respectively) (Figure 1). Therefore, we built three separate multivariable models including either the leptin level, serum albumin, or the MNA score.

Given the strong observed correlation between the CIRS-G and the ADL scores (Rho = −0.47, *p* < 0.0001), the latter variable was not included in the multivariate models. Similarly, the peripheral memory CD8 T-cell count was not considered further because it was correlated with the naïve memory CD8 T-cell count. In multivariate analyses (Table 4), Model 1 showed that a lower leptin level was significantly associated with HAIs, independently of the immune status (as assessed by the naïve CD8+ T-cell count), the CIRS-G score, and invasive procedures. Similar results were observed for a lower albumin value (Model 2) or an MNA score <17 (Model 3). After adjustment for the CIRS-G score or the invasive procedure, the CRP level was no longer associated with HAIs. Lastly, because of its association with the albumin level (Rho = −0.36, *p* < 0.001), the CIRS-G score was not associated with HAIs (Model 2). All of the three models had good discriminant ability (c-index: model 1, 0.79 (0.76–0.82); model 2, 0.81 (0.78–0.84); model 3, 0.80 (0.77–0.84)) and good calibration (0.35, 0.50, and 0.15, respectively).

Sensitivity analysis with dichotomized leptin levels (≤1st quartile vs. >1st quartile) produced very similar results (Appendix A).

## 4. Discussion

In a population of older patients hospitalized in a geriatric rehabilitation unit, the leptin level (i) was significantly higher in women than in men, (ii) was positively correlated with albumin in women only, and (iii) decreased significantly with increasing malnutrition class (MNA score) and decreasing BMI in both sexes. A lower leptin level was significantly associated with subsequent HAI in women only. The relationship between the leptin level and the occurrence of HAIs was abolished after adjustment for the MNA score or serum albumin. Three multivariate models showed that a lower leptin level, lower albumin level, and MNA score < 17 remained significantly associated with subsequent HAIs, independently of the CIRS-G score, invasive procedures, and immune status.

Our most striking finding was the association between baseline leptin levels and the occurrence of HAIs in women, independently of comorbidity, invasive procedures, and T cell counts. The present clinical study is the first to have shown that the leptin level is related to increased susceptibility to HAIs. Only a few studies have investigated the relationships between leptin levels and infection/sepsis in humans [11,27,28]. They found that the patient’s leptin level was low at the time of the infection and was correlated with the severity of sepsis [29,30,31,32,33]. However, these phenomena might have been related to metabolic and nutritional impairments caused by the infection, and so did not suggest that the leptin level is predictive of the occurrence of infections. Several studies have found a beneficial effect (decreased severity and cytokine production) of leptin administration in murine models of infection [34,35,36].

Leptin secretion is proportional to the adipocyte mass [37] and to malnutrition results in hypoleptinemia. Accordingly, a low leptin level has often (but not always) been associated with a low BMI and a low MNA score in adults with malnutrition [3,4,5,6,7], as was the case in our study population. In contrast to a previous study of adults having undergone colorectal cancer surgery, we did not find that a low BMI was associated with HAI [38]. Our observation suggests that the association linking leptin to HAI is probably not mediated by fat mass. The link between leptin and HAIs is probably more complex, as leptin has been shown to be a part of the intricate network that links nutrition, metabolism, and immunity [39,40]. The facts that (i) the MNA score and albuminemia were also strongly associated with subsequent HAIs and (ii) the association linking leptin to subsequent HAIs was no longer significant after adjustment for the MNA score or albumin suggest that the association between leptin and HAI is mediated by nutritional status. The potential relevance of sarcopenic obesity should be considered. Indeed, the risk factors of sarcopenic obesity include aging, malnutrition, sedentary lifestyle, hormonal deficiencies, and other molecular changes [41]. The majority of our population study presents these risk factors. Indeed, the muscle−fat imbalance with increasing age results in an increase in the pro-inflammatory adipokines secreted by adipocytes and the decline in the anti-inflammatory myokines secreted by myocytes, promoting a chronic low-grade inflammatory state.

With a view to identifying older patients at risk of HAIs in clinical practice, the MNA score and serum albumin might be easier and cheaper to measure than serum leptin. In contrast, the association between leptin and HAIs was maintained after adjustment for CRP. This suggests that inflammation (as evidenced by CRP levels in older patients with comorbidities) does not explain the link between leptin and subsequent HAIs. However other proinflammatory factors might be involved in the relationship with leptine (such as adiponectin, serum amyloid A, and osteopontin) [42]. More comprehensive studies are needed to explore the contribution of other factors/adipokines.

The leptin receptor is expressed throughout the immune system, and leptin has been shown to regulate both innate and adaptive immune responses in both humans and rodents [11,43,44,45,46,47]. The best-described immune consequences of leptin deficiency are changes in T-cell counts and functions [9,34,43,44,47]. As we reported previously [30], patients with vs. without HAIs have different T-cell profiles. We therefore analysed the T-mediated adaptive immunity in our study participants and found that the association between the serum leptin level and HAIs did not depend on the T-cell profile. Hence, the nature of the direct link between leptin and immune dysfunction is still not clear. Indeed, most studies of the link between leptin and immune dysfunction were performed in mouse models or on human immune cells ex vivo; various effects on adaptative and innate immune markers and cytokine production were observed [9,34,35,36,43,44,45,46,47]. In humans, only two cohorts with very specific leptin deficiency syndromes (congenital leptin deficiency (n = 3 children) in one [9] and hypothalamic amenorrhea (n = 14 women) in the other [47]) have been studied; in both cases, the abnormally low absolute CD4 T-cell count increased after treatment with leptin. However, these findings do not provide a mechanistic explanation for why the CD4 T cell count was low in patients with low leptin levels. The researchers referred to leptin’s concerted actions on the stress hormones or the circulating cytokines that control the survival of T cells, rather than a direct impact on CD4 T cells per se [9]. In the present study, the leptin level was associated with the CD4 count in men only, and so this finding requires confirmation. Lastly, the fact that the leptin level was associated with the occurrence of HAI in our study (independently of the T-cell profile) does not imply that this effect is independent of any other immune dysfunction; specific functional studies of this topic are warranted.

It is not clear why the association between leptin and HAI described here was only present in women. In the various above-mentioned preclinical and clinical studies of leptin level and infections, no sex differences were reported. The role of sex in the association between leptin and HAI is not clear and requires further investigation.

Our study had several limitations. Firstly, the single-centre study design and the small number of male patients might limit the external validity of our findings. Indeed, our results may only be relevant to this specific population of older patients in a geriatric rehabilitation unit presenting with comorbidities, disability, and a somewhat poor prognosis; 3 months later, around 44% of the patients had been discharged to an institution, were still in hospital, or had died. Secondly, nutritional status was assessed with the MNA score, rather than the more specific Global Leadership Initiative on Malnutrition definition [48]. However, an accurate history of weight loss (the most important phenotypic criterion) was not available in our study. We therefore completed the full MNA for all patients upon admission. The validated MNA has been widely used in various populations of older adults, including those in rehabilitation settings [23,49,50,51]. Thirdly, it would have been interesting to measure body composition, in order to better determine the role of fat and fat-free mass on the occurrence of HAI. However, bioelectrical impedance analysis and dual-energy X-ray absorptiometry were not readily available in our institution at the time of the study. Lastly, our immunological evaluation was limited in scope and examined the T-cell profile only; more comprehensive immunoprofiling and functional assays are now required.

## 5. Conclusions

Our present findings suggest that the low leptin levels observed in older women hospitalized in a rehabilitation ward contribute to increased susceptibility to infection. The relationship between leptin and subsequent HAI might be mediated by nutritional variables. The results of this exploratory study require confirmation by further research. Further investigations should include a body composition assessment and in-depth immunological measurements in a larger population of men and women.

## Figures and Tables

**Figure 1 nutrients-14-00226-f001:**
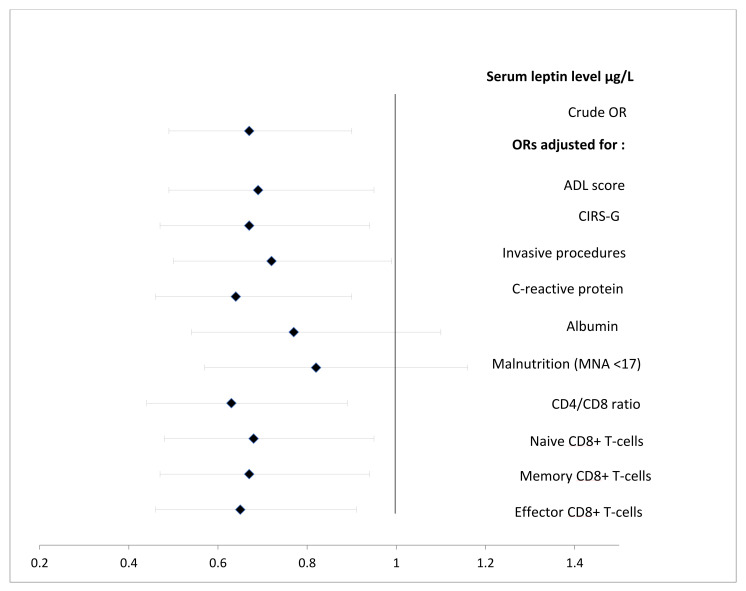
Adjustments for the ADL and CIRS-G scores; invasive procedures; serum CRP; serum albumin; the CD4/CD8 ratio; and the naïve CD8 T-cell, memory CD8 T-cell and effectors CD8 T-cell counts did not substantially change the relationship between the leptin level and the occurrence of HAIs.

**Table 1 nutrients-14-00226-t001:** Sociodemographic, nutritional, and laboratory characteristics of 232 patients hospitalized in a geriatric rehabilitation unit.

	MD	Total Population (N = 232)	Women (N = 168)	Men (N = 64)
Baseline characteristics				
Age (years)		84.8 [81.2–90.1]	84.5 [81–90]	84 [79–87]
ADL score		8 [4–12]	8 [5–12]	8 [3.5–11]
CIRS-G score		11 [9–14]	11 [9–14]	13 [10–15]
Invasive procedures		70 (30.2)	27 (42.2)	43 (25.6)
MMSE score	6	21 [16–25]	21 [16–25]	21 [17–26]
Body mass index (kg/m^2^)				
<22		78 (33.6)	62 (36.9)	16 (25)
22–29.9		112 (48.3)	74 (44.05)	38 (59.4)
≥30		42 (18.1)	32 (19.05)	10 (15.6)
MNA				
Malnutrition, <17		50 (21.6)	38 (22.6)	12 (18.8)
At risk, 17–23.5		125 (53.9)	90 (53.6)	35 (54.7)
Normal, 24–30		57 (24.6)	40 (23.8)	17 (26.6)
Serum albumin (g/L)	1	33.9 [30.9–37.4]	34 [30.9–37.3]	33.7 [30.8–37.5]
Serum C-reactive protein (mg/L)	1	6 [2.5–17]	5 [2.5–13]	8.5 [2.5–31]
Serum leptin (μg/L)		9.3 [3.7–21.6]	10.5 [4–24.4]	7.3 [2.5–12.6]
Immunological factors				
CD4/CD8 T-cell ratio	7	2.5 [1.4–3.5]	2.4 [1.4–3.5]	2.7 [1.4–3.5]
*CD4+ T-cells*	7			
Naïve CD45RA+CD62L+ (%)		17.5 [13.3–22.9]	17.1 [12.9–22.1]	18.8 [15–24.5]
*CD8+ T-cells*				
Naïve CD45RA+CD62L+ (%)	7	5 [3.3–7.3]	5.1 [3.5–7.1]	4.7 [3–7.8]
Peripheral memory CD45RA−CD62L− (%)	7	2.7 [1.5–5]	2.6 [1.4–5.4]	2.9 [1.8–4.5]
Terminal effector CD28− (%)	8	56 [38.5–69]	56 [37–69]	55 [42–66]
3-month outcome Occurrence of a healthcare-associated infection		89 (38.4)	63 (37.5)	26 (40.6)
Discharge to home		127 (55.7)	36 (57.1)	91 (55.2)
Discharge to an institution		57 (25.0)	14 (22.2)	43 (26.1)
Still in hospital	-	30 (13.2)	7 (11.1)	23 (13.9)
Deceased	-	14 (6.1)	6 (9.5)	8 (4.9)

MD—missing data; ADL—activities of daily living; CIRS-G—Cumulative Illness Rating Scale, Geriatric; MMSE—Mini-Mental State Examination; MNA—mini nutritional assessment. The data are quoted as the number (%) or the median [interquartile range].

**Table 2 nutrients-14-00226-t002:** Baseline characteristics and outcomes associated with the leptin level in the study population (n = 232).

		Leptin Level (μg/L)								
		Women (N = 168)			Men (N = 64)			Total Population (N = 232)		
		10.6 [4–24.4]			7.3 [2.5–12.6]			9.3 [3.7–21.6]		
		Spearman’s rho		*p* ^a^	Spearman’s rho		*p* ^a^	Spearman’s rho		*p* ^b^
Baseline characteristics										
Age (years)		−0.15		0.048	−0.09		0.50			
Serum albumin (g/L)		0.20		0.01	−0.02		0.89			
Serum C reactive protein (mg/L)		0.10		0.20	−0.15		0.25			
CD4/CD8 T-cell ratio		−0.102		0.20	−0.015		0.91	−0.9		0.16
*CD4+ T-cells*										
Naïve (CD45RA+CD62L+) (%)		−0.019		0.81	0.27		0.034			
*CD8+ T-cells*										
Naïve (CD45RA+CD62L+) (%)		0.08		0.28	−0.013		0.92	0.08		0.23
Peripheral memory (CD45RA−CD62L−) (%)		−0.028		0.72	0.04		0.78	−0.008		0.91
Terminal effector CD28− (%)		0.15		0.06	−0.13		0.31	0.09		0.20
		N	Median [IQR]	*p* ^a^	N	Median [IQR]	*p* ^a^	N	Median [IQR]	*p* ^b^
Body mass index	<22	62	3.5 [2.1–5.6]	<0.000	16	2.5 [1.1–4.8]	<0.000		3.3 [1.6–5.1]	<0.000
	22–29.9	74	13.9 [8.6–25.2]		38	7.5 [2.8–11.9]			11 [5.1–21]	
	≥30	32	45.9 [21.7–72.6]		10	21.4 [12.9–48]			38.1 [20.4–66]	
MNA	<17	38	3.8 [1.9–8.5]	<0.000	12	2.4 [0.95–6.3]	<0.000	50	3.5 [1.5–7.1]	<0.000
	17–23.5	90	10.7 [4.2–25.2]		35	5.7 [2.4–12.4]		125	9.5 [4–22.1]	
	24–30	40	21.8 [10.3–47]		17	12.4 [9.2–15]		57	16.4 [9.3–33]	
Outcome										
Healthcare-associated infection	No	105	11.8 [4.6–26.3]	0.01	38	6.4 [2.5–11.9]	0.35			
	Yes	63	5.9 [2.6–17.7]		26	7.9 [2.4–12.9]				

MNA—Mini Nutritional Assessment; median [IQR], median [interquartile range]. ^a^ *p* value from Spearman’s rank correlation test or the Kruskal−Wallis test, as appropriate, ^b^ *p* value for the quantile regression models, adjusted for sex.

**Table 3 nutrients-14-00226-t003:** Univariate analyses of women with vs. without one of more healthcare-associated infections (n = 168).

	HAI+	HAI−	Univariate Analysis	
	N = 63	N = 105	*p* ^a^	OR [95%CI] ^b^
Age (years)	85.5 [82–89]	85 [81–90]	0.78	-
ADL	7 [3–10]	9 [6–12]	0.001	0.88 [0.82–0.95]
CIRS-G^c^	13 [10–16]	10 [8–12]	<0.000	2.02 [1.39–2.94]
Invasive procedure	39 (52.9)	11 (9.7)	<0.000	10.43 [4.77–22.83]
Body mass index (kg/m^2^)			0.29	
<22	28 (44.4)	34 (32.4)		1.61 [0.81–3.23]
22–29.9	25 (39.7)	49 (46.7)		Ref.
≥30	10 (15.9)	22 (21)		0.90 [0.37–2.17]
MNA			0.001	
<17	24 (38.1)	14 (13.3)		5.90 [2.19–15.93]
17–23.5	30 (47.6)	60 (57.1)		1.72 [0.73–4.08]
24–30	9 (14.3)	31 (29.5)		Ref
Albumin ^d^ (g/L)	31.6 [28.7–34.1]	35.7 [31.9–37.8]	<0.000	0.39 [0.25–0.59]
C-reactive protein (mg/L) ^c^	8 [2.5–20]	2.5 [2.4–11]	0.01	1.65 [1.20–2.28]
CD4/CD8 ratio ^d^	2.0 [1.29–3.16]	2.56 [1.55–3.77]	0.07	0.75 [0.54–1.04]
Naïve CD4+ T-cells (CD45RA+CD62L+) (%)	16.5 [12.7–23.1]	17.25 [13.2–21.6]	0.53	-
*CD8+ T-cells*				
Naïve (CD45RA+CD62L+) (%) ^d^	4.8 [3.2–7]	5.25 [3.95–7.55]	0.05	0.69 [0.49–0.96]
Peripheral memory (CD45RA−CD62L−) (%) ^c^	2.9 [1.5–10.4]	2.35 [1.1–4.65]	0.03	1.53 [1.07–2.18]
Terminal effector CD28− (%) ^c^	62.5 [30–73]	52 [37.5–66]	0.09	1.09 [0.89–1.50]
Leptin level (μg/L) ^d^	5.9 [2.6–17.7]	11.8 [4.6–26.3]	0.01	0.67 [0.49–0.90]

HAI—healthcare-associated infection; OR—odds ratio; CI—confidence interval; ADL—activities of daily living; CIRS-G—Cumulative Illness Rating Scale, Geriatric; MNA—mini nutritional assessment. The data are quoted as the number (%) or median [interquartile range]. ^a^ *p* value from a chi-squared test or the Kruskal−Wallis test, as appropriate; ^b^ odds ratios (95%CI) were estimated using logistic regression models; ^c^ The odds ratios (95%CI) estimated using logistic regression models are quoted for a 1-SD increment in the log transformed values or ^d^ a 1-SD decrease in the log transformed values.

**Table 4 nutrients-14-00226-t004:** Multivariate analyses of women with at least one healthcare-associated infection vs. women without infections (n = 168).

	Model 1		Model 2		Model 3	
	OR [95%CI]	*p* Value	OR [95%CI]	*p* Value	OR [95%CI]	*p* Value
CIRS-G ^a^	1.58 [1.04–2.39]	0.03			1.54 [1.01–2.34]	0.044
Invasive procedure	6.30 [2.66–14.92]	<0.001	6.44 [2.73–15.20]	<0.001	6.23 [2.61–14.87]	<0.001
Naïve CD8+ T-cells (CD45RA+CD62L+)% ^b^	0.68 [0.46–0.99]	0.043	0.70 [0.49–1.03]	0.07	0.64 [0.43–0.94]	0.02
Serum leptin level (μg/L) ^b^	0.70 [0.49–0.97]	0.048				
Albumin			0.46 [0.29–0.74]	0.001		
MNA < 17					3.22 [1.36–7.63]	0.008

OR—odds ratio; CI—confidence interval; CIRS-G—Cumulative Illness Rating Scale, Geriatric; MNA—Mini Nutritional Assessment. The multivariate analysis was adjusted for all variables of the model. ^a^ The odds ratios (95%CI) estimated using logistic regression models are quoted for a 1-SD increment in the log transformed values or ^b^ a 1-SD decrease in the log transformed values.

## Data Availability

Some or all datasets generated during and/or analysed during the current study are not publicly available, but are available from the corresponding author upon reasonable request.

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
