# Peer review of "Serum Leptin Levels, Nutritional Status, and the Risk of Healthcare-Associated Infections in Hospitalized Older Adults"

_nutrients, 2022, doi:10.3390/nu14010226_

Round 1
Reviewer 1 Report
GENERAL COMMENTS
The manuscript addresses a topic of scientific interest, which is within the journal’s scope. However, some of the conclusions drawn from the data obtained may be difficult to follow. Clarification might be necessary and some aspects might be implemented, especially as regards contemplating other pathophysiological explanations. A broader view of potential underlying mechanisms might be interesting to comment.
The manuscript may benefit from considering the following aspects:
Abstract: indicate what is the true contribution of nutritional status to HAIs in the conclusions (currently only expressed as “may be opartly mediated by nutritional status”); the authors mention that leptin was not significantly associated with HAIs after adjustments for malnutrition or albuminemia.
Materials & Methods and throughout the manuscript, when referring to age suggest to include “years”.
Tables: indicate units of BMI = kg/m2
Discussion, page 10: as mentioned in the Abstract indicate what is the true contribution of nutritional status to HAIs in the conclusions (currently only expressed as “may be opartly mediated by nutritional status”); the authors mention that leptin was not significantly associated with HAIs after adjustments for malnutrition or albuminemia.
Given the advanced age of the study participants, the potential relevance of sarcopenic obesity should be contemplated (ref Evans K, Abdelhafiz D, Abdelhafiz AH. Sarcopenic obesity as a determinant of cardiovascular disease risk in older people: a systematic review. Postgrad Med. 2021 Nov;133(8):831-842). Risk factors of sarcopenic obesity include aging, malnutrition, sedentary lifestyle, hormonal deficiencies and other molecular changes. The muscle-fat imbalance with increasing age results in an increase in the pro-inflammatory adipokines secreted by adipocytes and a decline in the anti-inflammatory myokines secreted by myocytes. This imbalance promotes and perpetuates a chronic low-grade inflammatory state that is characteristic of sarcopenic obesity.
Discussion, page 10: the same applies to the conclusion pertaining to inflammation; the true contribution; the fact that the association was maintained after adjustment for CRP does not exclude that other proinflammatory factors might be involved in the relation with leptin (like adiponectin, serum amyloid A, osteopontin, etc.). In this line, it may be worthwhile mentioning that the involvement of other factors/adipokines can not discarded (ref Frühbeck G, Gómez-Ambrosi J. Rationale for the existence of additional adipostatic hormones. FASEB J. 2001 Sep;15(11):1996-2006.).
References #1 suggest a more recent review of leptin like Friedman JM. Leptin and the endocrine control of energy balance. Nat Metab. 2019 Aug;1(8):754-764.
Author Response
POINT-TO-POINT RESPONSES TO REVIEWERS
Nutrients
Manuscript reference: nutrients-1507179
Title: Serum leptin levels, nutritional status, and the risk of healthcare associated infections in hospitalized older adults
Article Type: Article
Author(s): Elena Paillaud * , Johanne Poisson , Clemence Granier , Antonin Ginguay , Anne Plonquet , Catherine Conti, Amaury Broussier, Agathe Raynaud-Simon , Sylvie Bastuji-Garin
RESPONSES TO REVIEWER #1
MINOR COMMENTS
- Abstract: indicate what is the true contribution of nutritional status to HAIs in the conclusions (currently only expressed as “may be opartly mediated by nutritional status”); the authors mention that leptin was not significantly associated with HAIs after adjustments for malnutrition or albuminemia.
Response: We thank the reviewers for this comment. It is indeed not clear and as leptin is associated to nutritional status we removed the may be partly from the abstract.
- Materials & Methods and throughout the manuscript, when referring to age suggest to include “years”.
Response: We added years when referring to age.
- Tables: indicate units of BMI = kg/m2
Response: We corrected it.
- Discussion, page 10: as mentioned in the Abstract indicate what is the true contribution of nutritional status to HAIs in the conclusions (currently only expressed as “may be opartly mediated by nutritional status”); the authors mention that leptin was not significantly associated with HAIs after adjustments for malnutrition or albuminemia.
Response: As in the abstract, we removed may be partly
- Given the advanced age of the study participants, the potential relevance of sarcopenic obesity should be contemplated (ref Evans K, Abdelhafiz D, Abdelhafiz AH. Sarcopenic obesity as a determinant of cardiovascular disease risk in older people: a systematic review. Postgrad Med. 2021 Nov;133(8):831-842). Risk factors of sarcopenic obesity include aging, malnutrition, sedentary lifestyle, hormonal deficiencies and other molecular changes. The muscle-fat imbalance with increasing age results in an increase in the pro-inflammatory adipokines secreted by adipocytes and a decline in the anti-inflammatory myokines secreted by myocytes. This imbalance promotes and perpetuates a chronic low-grade inflammatory state that is characteristic of sarcopenic obesity.
Response: We thanks the reviewer for this comments that enriches our discussion. We added the comment regarding sarcopenia and the reference (line 356-354).
- Discussion, page 10: the same applies to the conclusion pertaining to inflammation; the true contribution; the fact that the association was maintained after adjustment for CRP does not exclude that other proinflammatory factors might be involved in the relation with leptin (like adiponectin, serum amyloid A, osteopontin, etc.). In this line, it may be worthwhile mentioning that the involvement of other factors/adipokines can not discarded (ref Frühbeck G, Gómez- Ambrosi J. Rationale for the existence of additional adipostatic hormones. FASEB J. 2001 Sep;15(11):1996-2006.).
Response: We thanks the reviewer for this comment and added the comment regarding other potential proinflammatory factors and the reference in the discussion (line 371-374).
- References #1 suggest a more recent review of leptin like Friedman JM. Leptin and the endocrine control of energy balance. Nat Metab. 2019 Aug;1(8):754-764.
Response: We changed the first reference with this more recent reference, which is indeed more appropriate.
Reviewer 2 Report
This manuscript reports data on the link between rerum leptin levels in hospitalized older patients, finding significative results in women only, and the risk of healthcare associate infections occurrence.
It is well written and the design is appropriate with data accurately analized and reported in tables.
I would address just few concerns to hopefully improve the manuscript and help the readers:
- Implement the introduction with more estensive description of the relationship between leptine and Immune system, as well as leptin and fat mass due to the important difference found between men (not significant results) and women (who show higher fat mass). Leptin can be considered an adipokine and its function related to the immune system should be deepened and better explained.
- row 112-116 should represent a separate paragraph with another heading (invasive procedures?) why is it merged with the nutritional assessment?
- row 133 (not included? might be better than counted)
- should row 192-196 be included in "data analysis" instead?
- I would also change the legend of tablet: too long and those information should be included in the text instead. Just restrict the legend to abbreviations and info on the unit measures, while invasive procedures are described in the text (you may cite the paragraph if needed to report). Same for data analysis, should not be reported in the legends.
- Appendix is needed with MNA, CIRS-G, ADL and MMSE scores
Author Response
POINT-TO-POINT RESPONSES TO REVIEWERS
Nutrients
Manuscript reference: nutrients-1507179
Title: Serum leptin levels, nutritional status, and the risk of healthcare associated infections in hospitalized older adults
Article Type: Article
Author(s): Elena Paillaud * , Johanne Poisson , Clemence Granier , Antonin Ginguay , Anne Plonquet , Catherine Conti, Amaury Broussier, Agathe Raynaud-Simon , Sylvie Bastuji-Garin
RESPONSES TO REVIEWER #2
MINOR COMMENTS
- Implement the introduction with more estensive description of the relationship between leptine and Immune system, as well as leptin and fat mass due to the important difference found between men (not significant results) and women (who show higher fat mass). Leptin can be considered an adipokine and its function related to the immune system should be deepened and better explained.
Response: We thank the reviewers for this relevant comment. We agree that the interaction between leptin and immune system need to be discuss extensively. We feel that the introduction would be less clear if we developed in detail these interactions in this section, thus we developed it in the discussion (line 375-402).
- Row 112-116 should represent a separate paragraph with another heading (invasive procedures?) why is it merged with the nutritional assessment?
Response: It was indeed not clear and following your comment we modified it in the separate paragraph.
- Row 133 (not included? might be better than counted)
Response: We corrected it.
- Should row 192-196 be included in "data analysis" instead?
Response: It is a relevant comment and we corrected it and included this part in the data analysis section.
- I would also change the legend of tablet: too long and those informations should be included in the text instead. Just restrict the legend to abbreviations and info on the unit measures, while invasive procedures are described in the text (you may cite the paragraph if needed to report). Same for data analysis, should not be reported in the legends
Response: We removed the description of invasive procedures in table legend to shorten them. However, regarding some data analyses description, we feel that it is needed to allow the reader to read and understand the table quickly without the having to read the text. If needed by the editorial board, we can also remove it.
- Appendix is needed with MNA, CIRS-G, ADL and MMSE scores
Response: We thank the reviewer for this comment. If we understand correctly the reviewer is asking to add the description of every scores in an appendix. However, these scores have not been modified from the original published version and are well described in the reference that are cited for each one of them. We think that adding an appendix with their description should not be necessary. If needed by the editorial board, we can add it.